# Footprints to singularity: A global population model explains late 20th century slow-down and predicts peak within ten years

Christopher Bystroff[ID]*

Dept of Biological Sciences, Dept of Computer Science, Rensselaer Polytechnic Institute, Troy, NY, United Sates of America

* bystrc@rpi.edu, bystroffc@gmail.com

**Data Availability Statement:** The this study used global human population data, which are freely available at https://www.worldometers.info/world-population/world-population-by-year/ All other

## Abstract

Projections of future global human population are traditionally made using birth/death trend extrapolations, but these methods ignore limits. Expressing humanity as a K-selected species whose numbers are limited by the global carrying capacity produces a different outlook. Population data for the second millennium up to the year 1970 was fit to a hyper-exponential growth equation, where the rate constant for growth itself grows exponentially due to growth of life-saving technology. The discrepancies between the projected growth and the actual population data since 1970 are accounted for by a decrease in the global carrying capacity due to ecosystem degradation. A system dynamics model that best fits recent population numbers suggests that the global biocapacity may already have been reduced to one-half of its historical value and global carrying capacity may be at its 1965 level and falling. Simulations suggest that population may soon peak or may have already peaked. Population projections depend strongly on the unknown fragility or robustness of the Earth's essential ecosystem services that affect agricultural production. Numbers for the 2020 global census were not available for this study.

## Introduction

Global human population has grown alarmingly in the 20th century, leading to speculation about the maximum number we will reach and when. Estimates of the human carrying capacity vary widely [1], as widely as the fields of study that address population: economics, demographics, history, system dynamics, ecology, sociology, archeology, and bioinformatics to name a few. Some see doom and gloom in our collective future [2, 3], and rational discourse is inhibited by a broad and multifaceted taboo on discussion of population in general [4]. So global population dynamics remains understudied and poorly understood. The field lacks detailed numerical models and objective scrutiny, settling for informal, subjective and descriptive models that are poor predictors of the true numbers that we would like to know.

This paper asserts two points. (1) Global human population growth does not fit the exponential growth equation that governs all living things under static growth conditions. Instead it fits a hyper-exponential function in which the growth rate itself grows exponentially [5]. The implication is that there is a quantity that governs the rate of growth, and that it is itself a

data, model parameters, and code are stored at OSFHome (https://osf.io/q427b/).

**Funding:** There was no specific support for this work.

**Competing interests:** The author has declared that no competing interests exist. Salary was provided by RPI, NIH and Grantham Foundation. The donors, Grantham Foundation and NIH, do not have any competing interest in this work.

growing quantity. *Technology*, broadly defined, is such a quantity, since it drives increases in human life expectancy [6]. Knowledge of technology grows exponentially because established knowledge enables and accelerates the emergence of new knowledge. (2) Obviously, the world is finite and population growth cannot go on forever, but within that truth it is uncertain whether population will top-out at the carrying capacity, exceed it, or crash to a lower level. The outcome depends on the nature of the carrying capacity. In this work, carrying capacity is defined as a *dynamic number of humans that cannot be exceeded*, consistent with some prior use [7], but other prior uses treat carrying capacity as a static number or a number that can be exceeded. A dynamic hard limit is consistent with the treatment of humans as a K-selected species with a food-supply-limited population [8]. The food supply is in turn impacted by environmental degradation, including climate change [9]. These two concepts, hyperexponential technology-driven growth and dynamic carrying capacity-limited population, are encoded in a new, predictive system dynamics model.

Past predictions of the future of human population range from qualitative to quantitative. Demographic transition theory [10] asserts that technological advances decrease the death rate first, then decrease the birth rate, leading ultimately to stability. United Nations-sponsored extrapolations of trends in birth and death rates, sometimes including a stochastic treatment of migration, predict a population peak around 2050 at around 9 billion followed by a slow decline [11]. An older, surprisingly simple mathematical model fits population to a hyperbola [12]. Although presented in a tongue-in-cheek manner, the hyperbolic "doomsday" model nonetheless correctly predicted the world population within 8% for another 40 years after it was published in 1960. However, these models contain no explicit global limits.

The ecological footprint literature sets a numerical limit to the sustainable human impact on the planet [2]. If the "footprint" is not maintained within sustainable limits, then an increase in the mortality rate ensues. However, this negative feedback loop is only informally described. A key concept in moving from an informal to a formal (*i.e.* numerical and predictive) model is the notion that nature renders essential services to humanity in proportion to nature. Herein, the *ecosphere* is defined as the portion of the global biocapacity that is not appropriated by humans and that renders the essential ecosystem services. The *humansphere* is defined as the portion of the biocapacity that has been taken from the *ecosphere* and does not yield these services. These two terms come from the footprint literature [13], but here they are conceived with a few subtle differences. The *ecosphere*, but not the *humansphere*, absorbs carbon dioxide from the atmosphere to the extent that it contains growing plants. Absorption of carbon dioxide by the *humansphere* does not count because carbon dioxide absorbed into food by agriculture is re-emitted by decomposition or after eating by respiration. Likewise, the *ecosphere* but not the *humansphere* regenerates fresh water, regenerates soil fertility, pollinates flowering plants, and stabilizes the climate roughly in proportion to its fraction of the Earth's biocapacity. It also provides numerous support services "behind the scenes", such as maintenance of the food web that supports fish stocks and maintenance of the habitats of pollinating insects. To the extent that these services are lost, land from the *ecosphere* passes to the *humansphere*.

Technology and its effect on population has been written about extensively. Some view technology as an outgrowth of increased population ("More people means more Isaac Newtons") [14, 15] and others see it as a driver of growth [16]. Among those who see technology as a causal agent, some see it as an intrinsic property of all life, growing as if it were a living thing [3]. The so-called "singularity" is viewed as a point where technological advancement escapes human control [17]. To formalize the concept and make it predictive and numerical, we must define technology and attach it to something that we can measure. Herein, technology is defined as the capacity to decrease the death rate and to increase the carrying capacity. As such, technology is attached to the population growth rate.

System dynamics (SD) models for world population dating back to the 1970's included explicit limits to growth using ecological and economic feedback loops, providing a means to reproduce technology-driven hyper-exponential growth and to forecast a downturn [18, 19]. A 2004 "World3" SD model predicted our population future under various policy scenarios [20]. Its "standard run" predicted a peak population between 9 and 14 billion happening between years 2075 and 2085, followed by a decline of 20 to 40% over the subsequent 50 years, depending on estimates for global arable land. Several other world models were developed in the wake of World3, each of which endeavored to make the system more complex, not simpler. Authors of some follow-up studies chose to subdivide humanity [21], for example by breaking populations into age groups or regional numbers. Others introduced complex resource management systems [22]. The perceived complexity of SD models may have impeded their widespread adoption.

Modeling humanity as a K-selected species with technological acceleration of growth and a proportional degradation of the food production system leads inevitably to a boom/bust outcome. Consider that, first, humanity cannot exceed the food supply; second, that the food supply depends on ecosystem services that are essential for their production; and third, that those natural systems are being destroyed by human growth. Positive followed by negative feedback all but assures an overshoot followed by a crash in population, not a high plateau as some have predicted, although this is still possible.

This paper presents the arguments for this pessimistic projection. A formal model is used to explore the space of population collapse in detail, and future projections are fine-tuned by optimizing the model parameters against past population numbers and other data. Concepts derived from or inspired by the footprint literature [2], the theorized technological "singularity" [3], and the biological view of humanity as a K-selected species [8], have led to the system dynamics model presented here (Fig 1). This is a "mind size"[23] SD world model, offered with the hope of making a complex system simpler, easier to understand and easier to teach, and to better understand the Earth's limits to growth and humanity's likely future trajectory.

## Results

This section describes the results of the model generation process. The model generation process itself is described in **Methods** for those who are interested. Here we discuss the model structure, substructure, feedback and its output.

### The World4 model

The simple model, called World4, consists of four stocks and four flows, forming two interacting binary subsystems. There are two stocks quantifying states of the global environment (*humansphere* versus *ecosphere*) and two stocks quantifying states of technological development (*knowledge* versus *ignorance*). Both are closed systems, which means there are no outside sources or sinks. The only source of domesticated land (*humansphere*) is wild land (*ecosphere*), and life-saving technologies (*knowledge*) can only save more lives to the extent that there is a non-zero death rate (*ignorance*). In this model, *knowledge* draws down the death rate, while *ignorance* is the death rate itself. *Knowledge* does double duty by also increasing the carrying capacity. Other forms of knowledge and technology are ignored. We'll get back to technology, but first let's discuss humanity.

### The environment subsystem

Humanity is represented by its ecological footprint within the Environment subsystem. The footprint is the amount of Earth's biocapacity appropriated for human use [13]. The Earth has

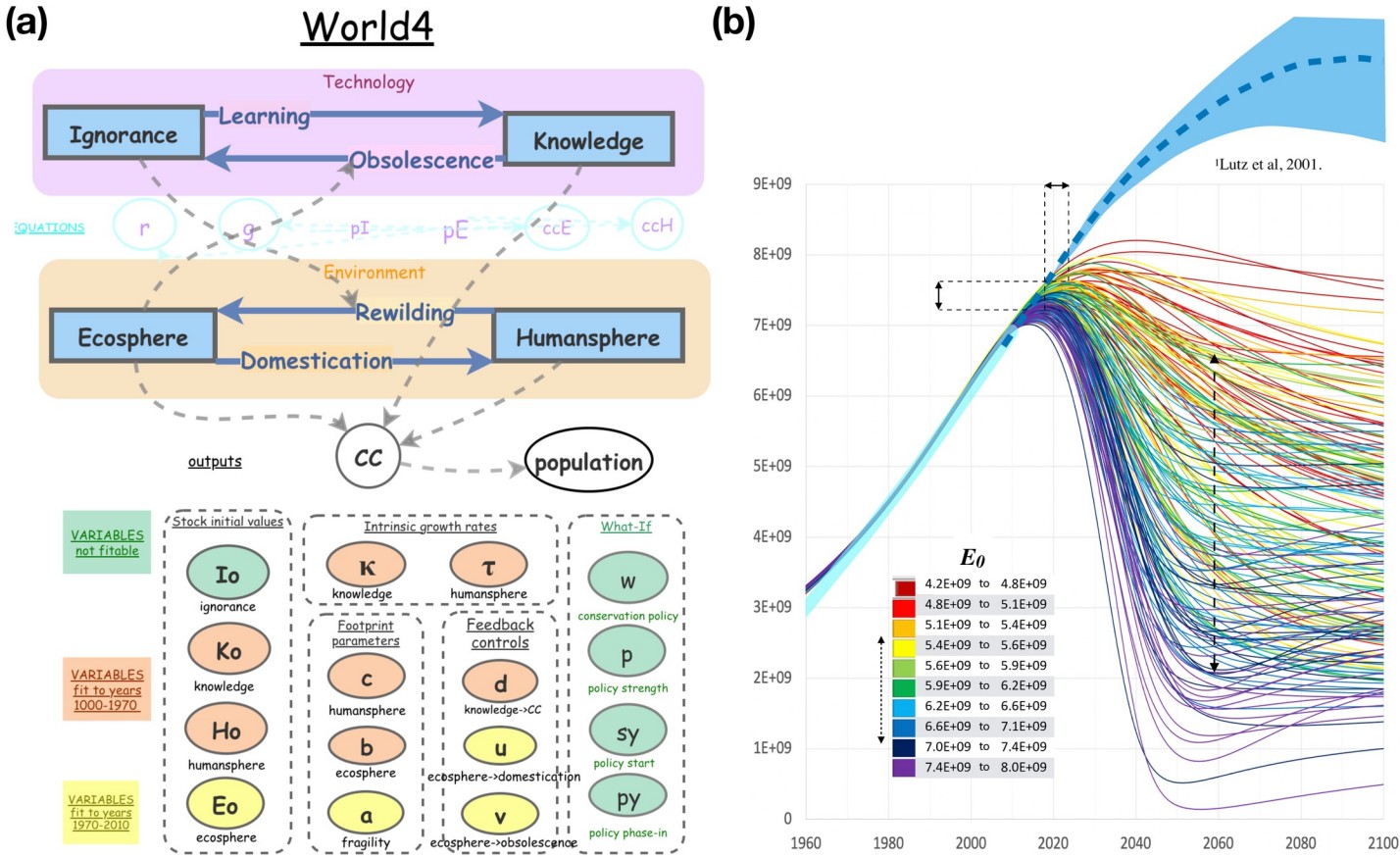

**Fig 1.** (a) World4, a system dynamics model that reproduces world population numbers up to 2010 and projects forward. Stocks (rectangles) and flows (solid arrows) form two interacting closed systems, one for Technology and one for Environment. Input variables (ovals) are colored and grouped by function. Output variables (white) are the global carrying capacity (*CC*) and *population*. Dashed lines indicate variable dependencies. (b) World4 simulations superposed on 20th century population numbers (thick cyan line) and UN population projections [11] (dashed blue line is the median projection and light blue are 95% confidence region). The program *hyperfit* carried out 1 million World4 simulations using randomly selected parameters from ranges listed in Table 1. Shown are the 184 trajectories that deviate from 1970–2010 population data by less than *rms* 0.5e8. Simulations are colored by their $E_0$ value (total ecosystem size in *gha*, see inset). Counterintuitively, a low $E_0$ means a higher population is sustainable. Double-headed arrows indicate 80% confidence ranges for peak date, peak height and 2060 population.

a maximum total biocapacity ($E_0$) estimated to be around 1.12e10 global hectares (gha) which is shared between wild and domesticated land and sea. *Humansphere* is equal to population times the average consumption per capita, times a term derived from the state of technology. In terms of the popular "I = PAT" [24] for ecological impact, *humansphere* is "I", *population* is "P", and the carrying capacity equation (*CC*) expresses the "A" (affluence) and "T" (technology) terms. In this model, *population* is viewed as a consequence rather than a cause of consumption. Therefore the expression becomes P = I/(AT), or

$$population = humansphere \cdot CC, \tag{1}$$

where *CC* is the reciprocal encoding of the informal concepts *A* and *T*. *Humansphere* grows intrinsically in the model, via the flow

$$domestication = (I_0 + \ln(2)/\tau)(1 - \exp(u\, ecosphere/E_0)) \cdot humansphere \tag{2}$$

because human need for more land grows with, and is proportional to, the population. The rate constant $(I_0 + \ln(2)/\tau)\,(1 - \exp(u\, ecosphere/E_0))$ approaches zero as *ecosphere* approaches zero because *ecosphere* cannot be negative. *u* is a negative number that models the strength of

this feedback. $I_0$ + ln(2)/$\tau$ is the birth rate, which is the replacement of deaths ($I_0$) plus the net growth, where the latter is reciprocally related to the intrinsic doubling time $\tau$, in years. The approach of basing population on carrying capacity is contrary to most population models, including World3 [19], that express population in terms of births and deaths. Hopfenberg [8] and others have argued that population growth should be viewed as food-supply driven rather than the result of births and deaths. This view has been met with skepticism [25] but it is consistent with the treatment of humans as a K-selected species. To treat humans differently from all other K-selected species would be a form of "human exceptionalism" [26], which is not scientific. Moreover, *ecosphere* cannot be expressed in terms of a population, so *gha* units make sense for these stocks.

## The technology subsystem

In this model, technology is a driver of growth through the suppression of mortality and is a factor in amplifying the human carrying capacity. In a sense, technology (or knowledge of technology) is an independent living entity since it can exist in written form outside of humanity itself and since it has its own catalytic effect on the development of new technology. Thus it is not wrong to treat technology as a living thing that can grow exponentially. The expression for intrinsic flow into *knowledge* is

$$learning = \kappa \cdot knowledge \qquad (3)$$

where $\kappa$ is a constant. Also, like a living thing, technology can "die" by obsolescence. Even though the knowledge may still remain, its usefulness towards human survival can whither to zero (buggy whip technology, for instance). Thus in the model we lump together unlearned or unknown technology with obsolescent or ineffective technology under the term *ignorance*, a quantity that is rate constant for mortality.

## Feedback between environment and technology subsystems

The two subsystems, Environment and Technology, interact to form a feedback loop. Specifically, since humanity is expressed in terms of its ecological footprint in *gha* units, and because death represents the return of human appropriated biocapacity to the *ecosphere*, therefore *ignorance* is the rate constant for that flow from *humansphere* to *ecosphere*, called *rewilding*.

$$rewilding = ignorance \cdot humansphere \qquad (4)$$

It is postulated that the loss of the undomesticated environment will cause technological challenges to human survival due to the loss of ecosystem services that are required for food production. To model this, *ecosphere* feeds back to *obsolescence* as follows.

$$obsolescence = 0.5 \exp(\nu \, ecosphere/E_0) \cdot knowledge \qquad (5)$$

where $\nu$ is a negative number. Eq 5 expresses the idea that as the wild environment disappears (depleting fresh water aquifers and fossil fuels, for instance), old technologies lose their usefulness (center-pivot groundwater irrigation systems, oil-fired electric generators), and humanity must develop new technologies to survive (ocean desalination, photovoltaic panels). Thus it makes sense that *ecosphere* depletion leads to *obsolescence*. This completes a negative feedback loop (Fig 2A). To play out a scenario on this loop, we can imagine that *ignorance* goes to zero, therefore *rewilding* decreases. This leads to a decrease in *ecosphere* by *domestication*. Depletion of *ecosphere* causes an increase in *ignorance*, therefore an increase in *rewilding*, reversing the effect. In the context of the exponentially depleting quantities *ignorance* and *ecosphere*, the system reaches a switching point and thereafter equilibrates (Fig 2B). Interestingly, this is the

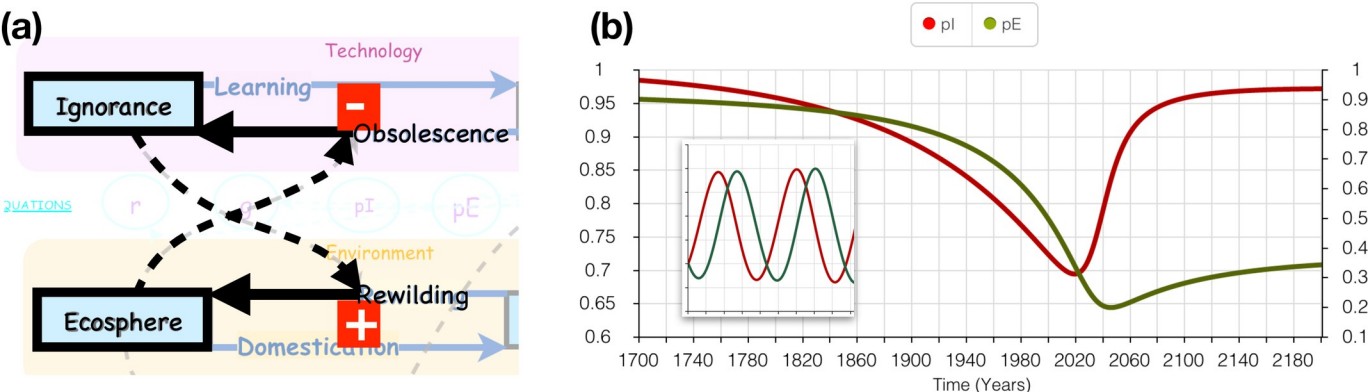

**Fig 2. Feedback.** (a) *Ignorance* feeds back in a positive way to *rewilding*. *Rewilding* increases *ecosphere*. *Ecosphere* feeds back negatively to *obsolescence*. *Obsolescence* increases *ignorance*. (b) Exponential decrease of both *ecosphere* (pE) by *domestication* and *ignorance* (pI) by *learning*, results in a switch, first in *ignorance* then in *ecosphere*. Inset: undamped Lotke-Volterra oscillation.

same feedback cycle that generates Lotke-Volterra oscillation (inset in Fig 2B) Interestingly, this feedback cycle generates Lotka-Volterra oscillation [27] (inset in Fig 2B) under different parameter setting. In the current setting, both *ecosphere* and *ignorance* are decreasing exponentially, causing the oscillation to be severely damped, producing just a single oscillation.

## Fitting the parameters of the model to population data

Parameter settings for the model were determined by non-linear least squares fitting of population output to historical population data, as described in greater detail in Methods. The intent of the project was to explain the hyper-exponential growth that was observed in the 19th and 20th centuries, to explain why recent population growth has been slowing, and to predict the future (Fig 3D). Fitting data before 1970 to a hyper-exponential model asks the question "*How did humanity grow in the absence of limits*?" Whereas, fitting the late 20th century slowing asks the question "*How do planetary limitations slow the growth of humanity*?" The result of the fitting is a model that projects into the future, providing a range of different population projections. Details of the process are found in Methods.

## Output of the model

After fitting, the output of this model matches all historical population data from years 1500 to 2010 within ±10% with the exception of the 1950 census, which is overestimated by 14%. It should be noted that 1950 census number was revised 17 times from 1951 to 1996, mostly upward [28]. Population growth has been sub-exponential over the last 50 years, suggesting that humanity is passing through an inflection point of a curve that is the product of two steep trends, one upward, the other downward. The upward curve is the combined exponential expansion of humanity and the intrinsically exponential increase in technological innovation, and the downward curve is the accelerating depletion of non-renewable resources and the loss of food security. The model predicts that the results of the 2020 census (not yet available) will be in the range 7.2 to 7.6 billion (80% confidence) instead of the projected 7.8 billion [11]. The model predicts that the population will peak or has peaked, with the peak in the range 2018 to 2023 (80% confidence) and will decline to between 2.1 and 6.6 billion by 2060. The nearness of the peak is supported by accelerating increases in adult mortality and decreases in birth rates since 2016 [29].

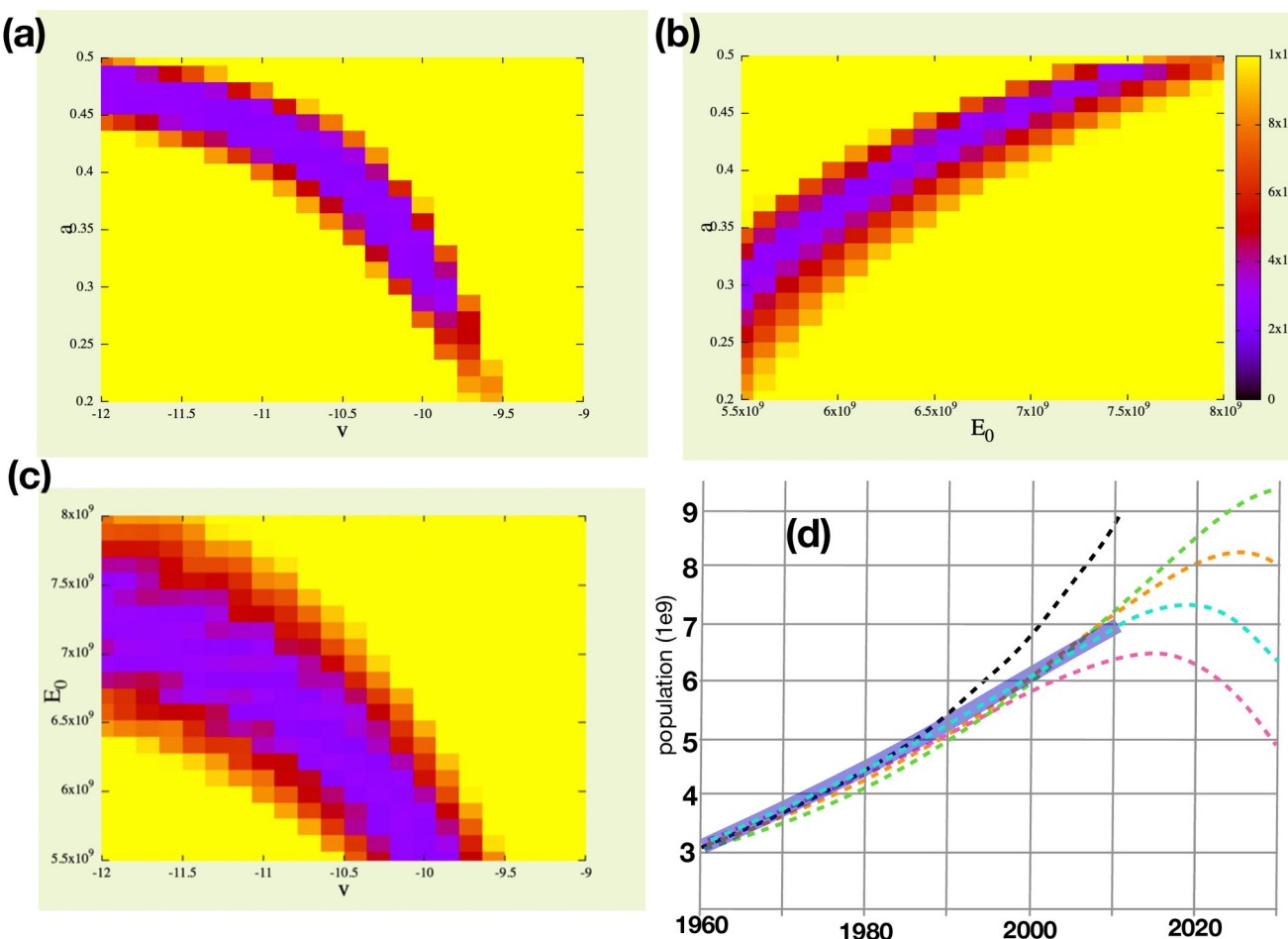

**Fig 3. Least-squares fits to years 1970–2010 are non-linearly correlated in the space of the four variables ($E_0$, $a$, $u$ and $v$) that effect only recent population data, as shown using *hyperfit*.** For example, as seen in (a), the best-fit setting for $v$ (ecosystem-dependent obsolescence of technology) goes down as we increase the setting for $a$ (ecosystem fragility). Each image is a projection of minimum values of residuals from the 4D space to 2D spaces (a) $a$, $v$, (b) $a$, $E_0$, (c) $E_0$, $v$. (d) A plot of five trajectories using optimal and suboptimal values, demonstrating the effect of choice of $E_0$ (total ecosystem size) on growth rate (1960–2000) and on the position of the population peak, ignoring other parameters. A hypothesized infinite ecosystem, $E_0 = \infty$ (black), leads to massive overestimate of growth rate. $E_0 = 0.800e10$ (green) or $E_0 = 0.750e10$ (orange) overestimates growth rate and predicts a later peak. $E_0 = 0.695e10$ (cyan) is optimal and predicts a 2020 peak. $E_0 = 0.600e10$ (magenta) underestimates growth and predicts that we are past the peak. Thick blue line is population data up to 2010. 2020 population numbers are not available.

## Significance of the model

The system dynamics simulation assumes fixed set of parameters and equations throughout the simulated time period. The model has no outside sources or sinks. There are no built-in switches and no settings are changed at any point in the simulation. Nonetheless, the simulation matches very closely to 2010 years of population history spanning periods of slow growth, rapid growth, and recent deceleration. The results of the study are a set of meaningful parameters that attach numbers to well-established but theoretical human/environment feedback mechanisms. Parametric solutions from the multi-variable least squares fit span only a narrow range in most variables, varying for the most part in the parameters that define the feedback between human population and the environment. Multiple solutions fit and explain past population data equally well but project very differently into the future as shown by the narrow 80% confidence ranges for the population peak and the side confidence range for the population in 2060 (see Fig 1B).

## Limitations of the model

The theoretical appearance of a population decline in the near future is a foregone conclusion of the design of the model itself. The model encodes the business-as-usual (BAU) assumption that humanity will not react to change and will continue to degrade the environment. It also assumes that the carrying capacity depends critically on resources that are not under human control nor are regenerated by human activity, and which will not come under human control within the timeframe of the simulation. These model design choices are not to be considered incontrovertible, but they are consistent with the dominant theory in human ecology and are intended to be free of human exceptionalism.

## Discussion

### Modeling philosophy and motive

Human population dynamics has its root causes in the environment and in human behavior. An understanding of the essence of our interactions with Nature and with each other can be best gained by expressing human population in the simplest possible dynamic model. The approach taken here has been to define the components and equations of a systems model and to tune the parameters of these components to fit population data. Prior to the current model we explored a number of alternative models or equations and rejected them, either because they could not be tuned to fit the data, or because they did not make real world sense, or because they were too complex. Because accurate prediction of future population was the motive from the outset, the simplest possible model was designed, giving the best chance of being robust to errors in the data and mistakes in the functional forms.

### Exploring parameter space

The behavior of the model with respect to its parameters is complex and surprising. For example, if $E_0$ is set to 7.50e9, it places the peak population year at 2038, but then the model fits poorly to the data, showing an upwardly curving trajectory through the 1980's and 90's (see Fig 3D), though we know that did not happen. Setting $E_0$ to a low value of 5.5e9 places the peak at 2016 and shows a downward curving population trajectory from 1990 to 2010, which again we know did not happen. But if all other parameters are allowed to float and are fit to the data by exhaustive multidimensional search, then both high and low settings of $E_0$ match very well to the 1970–2010 data (see Fig 1B). However, the results are reversed. The peak for low $E_0$ is now at 2040, and the peak for high $E_0$ is now at 2016. Surprisingly, in the context of the full complement of model parameters, the effect of $E_0$ has become completely counter-intuitive. With a little effort we can rationalize this strange behavior, as follows. If $E_0$ is high (8.0e9 gha) and yet population data is well fit, the parameter $a$ adopts a high value ($a$ = 0.48), which means the ecosystem is fragile and fails well before it is completely depleted. This leads to a steeper downturn in carrying capacity, which leads to the observed prediction, a counterintuitive earlier peak. On the other hand, if $E_0$ is set to a low value ($E_0$ = 4.2e9) and yet population data is well fit, then the parameter $a$ adopts a low value ($a$ = 0.10) and a later population peak is predicted. Low $a$ is interpreted to mean that the ecosystem is robust and the ecosystem services upon which we all depend are not entirely provided by the wild *ecosphere* but may be partially provided by the *humansphere*. To be clear, these very different predicted futures are not scenarios that can be affected by policy change, rather they are the results of uncertainty and a lack of understanding of the fragility or robustness of the global ecosystem with respect to human carrying capacity. Upcoming results from the 2020 census will greatly resolve this uncertainty.

## Analysis of the median business-as-usual projection

The World4 model is a BAU projection in the sense that human growth behavior and behavior towards the environment are treated as parameters that are determined by fitting past population data, not parameters that in any way seek an outcome. A range of parametric solutions has been found by the process described here. Each solution fits past population data the same but projects differently into the future. The median projection (Fig 1) peaks at or around 2022, then falls steeply, leveling off at around 3 billion by 2060. The cause of the decline within the model is a decrease in the food supply caused in turn by degradation of the environment and the concomitant atenuation of essential ecosystem services. The model reflects the current thinking on climate change and its repercussions. Climate change leads to weather uncertainty, increased severe storms, draught and floods, and sea level rise affecting low-lying areas —each a factor in decreasing agricultural output. Increased hunger in turn fuels conflict [30, 31]. Conflict leads to further decreases in food production and to mass migration, as we have seen recently from the rapidly heating Sahel region of Africa [32, 33]. In the BAU projection we see an increase in human mortality, followed by a decrease in carrying capacity. The recent worldwide spread of Covid-19 is an example of an emerging source of mortality for which human technology was not ready. Societal stressors such as hunger or a pandemic can drive violent behavior [34]. In the medan projection, following the peak, population drops quickly, accelerating to 100 million net lives lost per year through the years 2030 to 2040, which is faster than the fastest growth during the 20th century. In this model, we clearly see the cause of the rapid decline—the exponential growth of the consumption of finite vital natural resources.

## Real world examples of current ecosphere collapse

We are beginning to see the effects of the decline of natural resources and ecosystem services. Fossil fuels, fresh water aquifers, and greenhouse gas sequestration by plants are all regarded in this model as part of the *ecosphere* since they are not generated by human activity. The decline of one or more natural resources is cited as a cause of, for example, the ongoing deadly conflicts in Syria starting since 2011, the conflicts and famine in Yemen beginning in 2015, and the economic collapse in Venezuela that began in 2014, to mention a few. Draught and desertification were blamed for conflicts and mass migration out of the Sahel region of Africa, where Lake Chad has all but disappeared. Conflicts and famine have produced millions of refugees. Innumerable lives have been lost crossing the Sahara or crossing the Mediterranean, or in primitive camps along the southern borders of Europe. The 2017 documentary film "Human Flow" [35] reveals the massive scale of the refugee issue. Meanwhile, the global north has responded to the aggregate changes of the last 50 years with a dramatic decrease in the birth rate. An increasingly technological workforce has meant women spend more time in school and marry later. Rising oil prices have steadily ramped up the cost of raising a child, leading to smaller families by choice. Total fertility rate (TFR) globally is projected to reach replacement level (2.11) this year, 2020.

## Real world examples of the predicted decline of technology

Along with ecosystem decline, the model predicts changes in technology. In the projection, knowledge will be lost or made obsolescent during a population collapse. Much of our cultural technology is composed of laws, governance and economics. In recent economic history, consumerism has become engrained in our culture [36, 37]. Stability and prosperity in the context of the current economic system relies on population growth, according to economists. A technology shift in economics is likely when population begins to decline, since growth-based economic systems and the associated body of knowledge will become obsolescent in the sense that

they will not produce stability. In effect, economics will have to be re-learned. Obsolescence of growth-based economics may manifest itself in real-world breakdown in economic systems leading to decreased efficiency in manufacture and trade, in turn leading to a decrease in the effective food supply, which in turn will cause an increase in malnutrition and a decreased birth rate. Already, increased adult mortality and decreased birth rate are both current trends in global vital statistics [38]. In other areas, medical technology is partially responsible for a historic low death rate worldwide, but successful treatment of disease requires instruments and drugs that depend on a complex supply chain and high level engineering skills. In the event of an economic shift, supply chains will be disrupted unless a new system of economic motivation is quickly invented to replace the growth motive. In agriculture, technology to increase crop yields will become obsolescent as climate challenges, biodiversity losses (especially the loss of pollinating insects), and depletion of freshwater aquifers, combine with economic changes to reduce the efficiency of food production and distribution.

### Hopeful what-if scenarios as add-ons to World4

The future could easily be different than what is predicted by World4. World4 does not model changes in attitude and policy. Humanity could adapt in ways that may be good or bad. Modeling adaptation mechanisms opens a non-BAU modeling space that is too large to thoroughly explore. Taking inspiration from E. O. Wilson's book "Half Earth" [39], parameters for policies to preserve wild nature were implemented. Four new variables were added, **w, y, py** and **sy**, as defined in Table 1, for the target amount of *ecosphere* to save, the level of policy enforcement, the phase-in period and the date on which the policy begins, respectively. These variables do not affect populations prior to and including 2010. Preserving wild land wild allows humans to thrive. The optimal result (coindidentally it is **w** = 0.5, half earth!) gives, as Wilson predicted, a stable and high human population (Fig 4). This makes sense mathematically, because the carrying capacity equation contains a term of the form *x(1-x)*, which has a maximum at x = 0.5 where x is the fraction of the Earth dedicated to the *ecosphere*. But it also makes sense ecologically, because maximum sustainable food production is a trade off between maximizing arable land and maximizing climate stability, the latter embodied in wild forests and arctic ice, and other buffers to change. A global climate awareness campaign might lead to such a balance.

Any attempt to halt growth has to address the population. Cultural taboos currently prevent discussing, much less solving, this problem [4]. But in a what-if scenario, we can imagine ways that population growth can be halted or even reversed while preserving peace and prosperity. To build a mental picture of a society that has achieved balance with nature, imagine a people with a strong religious prohibition against growth, so engrained that no policing is required. A woman of child-bearing age in the Half-Earth world are permitted to have another child only if she is "blessed" by an elderly person, who, on his deathbed, bequeaths to her his one and only "blessing"—the right to procreate. The one-to-one matching of deaths to births would guarantee population stability.

### Conclusions

Hyperexponential population growth implies a dynamic system with two intrinsically growing quantities, human impact and technology. In a closed system, growth of one quantity implies depletion of another. *Knowledge* depletes *ignorance*, and the *humansphere* depletes the *ecosphere*. The quantities that are being depleted affect each other in a negative feedback loop leading to a sharp peak followed by a collapse of *humansphere* and *knowledge*. The timing of the collapse was determined by fitting the global limits to population in the context of

**Table 1. Complete component list for World4 model in four parts.** (a) Variables. (b) Flows. (c) Stocks. (d) Equations. Variables in bold italics were fit to data. Best value is one solution of many. Range shows values that can be fit to data with less than a specific residual depending on range of years fit. Fit years is the range used for fitting in *hyperfit*.

**(a)**

| Var. | Best value | Range | Fit years | Physical meaning |
|---|---|---|---|---|
| *a* | 0.426 | 0.35 to 0.48 | 1970–2010 | Ecosystem fragility. Relates $cc_E$ vs *ecosphere*. A higher/lower *a* means that ecosystem services are fragile/robust with respect to *ecosphere*, respectively |
| *B* | 1.0 people/gha | 0.7 to 1.7 | 1000–1970 | Base level carrying capacity for *ecosphere*. |
| *C* | 5.5 people/gha | 4.5 to 7.0 | 1000–1970 | Base level carrying capacity for *humansphere*. |
| *D* | -110 | -150 to -90 | n/a | Rule of diminishing returns. Relates *knowledge* to *CC*. A more negative value for *d* means *knowledge* raises *CC* more. |
| $E_0$ | 7.05E+09 gha | 4.3e9 to 8.1e9 | 1970–2010 | Initial biocapacity of the *ecosphere*. |
| $H_0$ | 1.5e8 gha | 1.2e8 to 1.6e8 | 1–1970 | Domesticated land in 0CE. Initial value of *humansphere*. |
| $I_0$ | 0.05 $y^{-1}$ | 0.05 to 0.25 | n/a | Base mortality. Multiplied by *humansphere* to get *rewilding*. Must be higher than maximum value of *knowledge*. Past population is insensitive to this variable but it affects future population. |
| $K_0$ | 7.25e-11 $y^{-1}$ | 2.0e-11 to 2.0e-9 | 1000–1970 | Technology in Year 0. Initial value of *knowledge* in 0CE. |
| *P* | n/a | 0 to 1.0 | n/a | Enforcement level of conservation policy. Higher *p* means stronger enforcement of policy. |
| *Py* | n/a | 0 to inf. | n/a | Policy phase-in time of conservation policy. Linear phase-in for enforcement of conservation policy *w*. |
| *Sy* | n/a | 1960 to inf. | n/a | Starting date of phase-in of conservation policy. When $p_E < w$, *domestication* is multiplied by g = g(((y-*sy*)/*py*)p+(1-(y-*sy*)/*py*)(1-exp(-10(w-$p_E$))) + exp(-10(w-$p_E$)), where y is the current year. Used only in the phase-in period *sy* through *sy*+*py*. |
| *u* | -8.6 | -inf. to -6.5 | 1970–2010 | Aggressiveness of growth. |
| *V* | -11.46 | -inf. to -9.0 | 1970–2010 | Aggressiveness of technological development. |
| *W* | n/a | 0 to 0.5 | n/a | Fraction of *ecosphere* to save using conservation policy. When $p_E < w$, *domestication* is multiplied by p (1-exp(-10(w-$p_E$))) + exp(-10(w-$p_E$)) |
| *K* | 9.6E-03 $y^{-1}$ | 6.5e-3 to 1.0e-2 | 1000–1970 | Learning rate. Rate of the intrinsic growth of *knowledge*. |
| *T* | 852 y | 700 to 1525 | 1–1970 | Doubling time of *humansphere* in Year 0. |

**(b)**

| Flow | Source | Sink | Formula | Physical meaning |
|---|---|---|---|---|
| *Rewilding* | *humansphere* | *ecosphere* | *ignorance*\**humansphere* | Deaths expressed as change in ecological footprint. |
| *domestication* | *ecosphere* | *humansphere* | *g*\**humansphere* | Births expressed as change in ecological footprint. |
| *Learning* | *ignorance* | *knowledge* | *κ*\**knowledge* | Intrinsic technology growth. |
| *obsolescence* | *knowledge* | *ignorance* | *r*\**knowledge* | Loss of technology. |

**(c)**

| Stock | Initial value | Physical meaning |
|---|---|---|
| *humansphere* | $H_0$ | Amount of total biocapacity appropriated for human use in Year 0, in gha. |
| *ecosphere* | $E_0$ | Amount of total biocapacity not appropriated for human use in Year 0, in gha. |
| *knowledge* | $K_0$ | Mortality eradicated by technology, in per year rate units $y^{-1}$. |
| *ignorance* | $I_0$ | Base mortality rate. Eradicated by technology. In per year rate units, $y^{-1}$. |

**(d)**

| Equation | Formula | Physical meaning |
|---|---|---|
| $cc_E$ | $\boldsymbol{b} \, p_E^{(0.5/(1+pE-2a))}$ | Carrying capacity contributed by the ecosphere. |
| $cc_H$ | *c* (1—exp(*d* \* *knowledge*)) $cc_E$ | Carrying capacity contributed by the humansphere. |
| $p_E$ | *ecosphere*/$E_0$ | The wild fraction of the environment. |
| *G* | ($I_0$ + ln(2)/*τ*)(1-exp(*u* $p_E$)) | ecosphere-dependent net intrinsic growth rate of humansphere |
| *R* | exp(*v* $p_E$) | ecosphere-dependent depletion rate of knowledge |

*(Continued)*

**Table 1.** (Continued)

| (a) | | | | |
|---|---|---|---|---|
| **Var.** | **Best value** | **Range** | **Fit years** | **Physical meaning** |
| **CC** | $cc_E + cc_H$ | Global carrying capacity in humans per gha. | | |
| **population** | $CC^*humansphere$ | Carrying capacity determines population number. | | |

hyperexponential growth. Population is predicted with 80% confidence to be 7.0 to 7.5 billion in 2020, and to peak between years 2018 and 2023 at a value of between 7.2 and 7.6 billion people. A much clearer picture will emerge when the 2020 census data is available.

## Methods

This section describes the conceptual generation of the model and the reasoning behind the equations within the model, as well as a detailed description of the fitting process.

### The hyper-exponential growth model explains most of human history

The hyper-exponential growth of human population from 1000 CE to the present is revealed by simple curve fitting against historical population data.

$$log\ P_t = \log(1.36e8) + 7.78e{-}4\ t + 8.10e{-}9\ \exp(9.74e{-}3\ t) \tag{6}$$

where $t$ is the year since 0 CE and $P_t$ is population at $t$. This formula fits all population numbers from 1700–1987 within 10%, and all numbers from the present back to 1000CE within

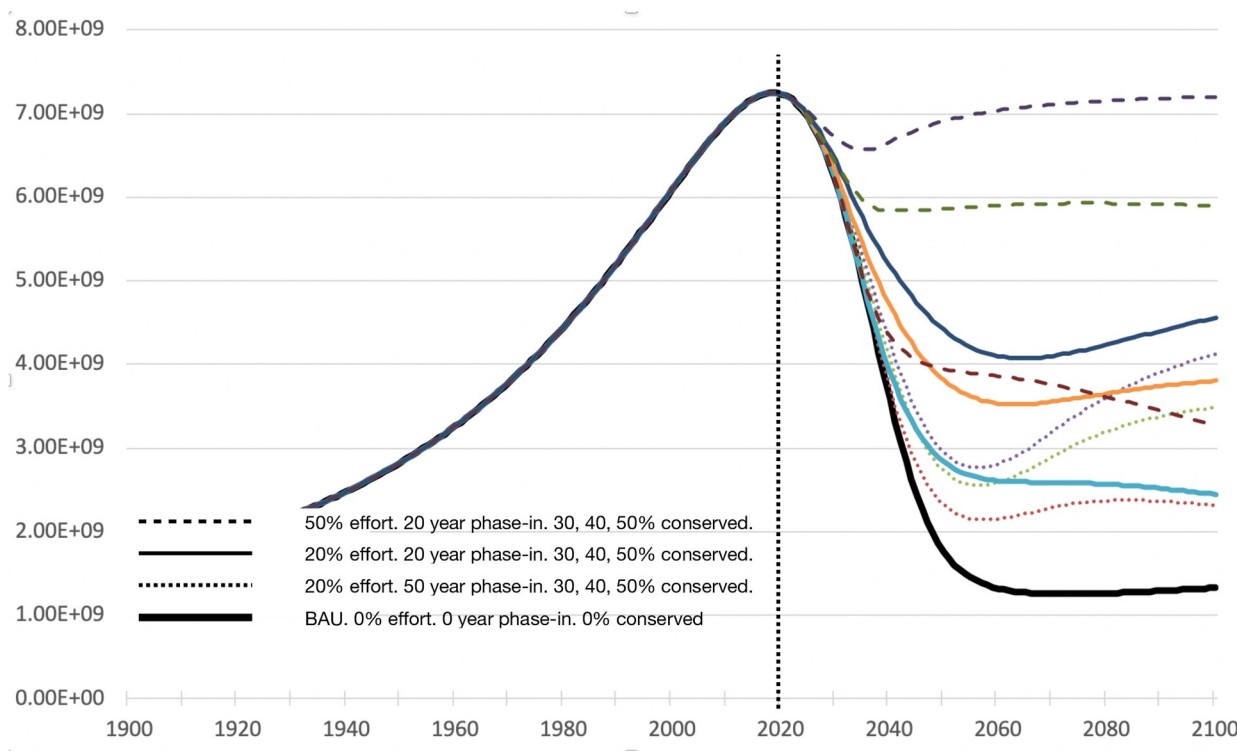

**Fig 4. Half-Earth scenarios.** Population projections for conservation efforts with various % enforcement, length of phase-in period and the % of ecosphere to be preserved, as compared to a BAU scenario. Dotted line is the present year, 2020.

17%, except 2010. Note that this equation differs from an earlier published one (Eq 12 in [5]) with the addition of a log-linear term (*7.78e-4 t*). This term improves the fit to numbers in the Renaissance period. Eq 6 may be interpreted as exponential growth in which the growth rate grows exponentially due to an exponential decline in the death rate. Other functional forms of the growth equation were considered and discarded, either because they did not fit the data or because they were not realistic. For example, normal exponential growth fits poorly to the data after 1500. On the other hand, a hyperbolic function fits all the data but lacks a physical rationale. Only the hyper-exponential equation achieves the steepness of human population growth in the 20th century without implying an unrealistic asymptote or arbitrary timepoints of discontinuity.

## Hyper-exponential growth equation implies a 2 subsystem model

The hyper-exponential equation has two intrinsic growth rates. Therefore it implies two interacting subsystems, one containing the human population and the other containing a quantity that affects the rate of human growth. For this quantity we use the term "*knowledge*", meaning knowledge of technology. Technology generally improves life expectancy and efficiency in the use of natural resources and thus accelerates growth [6]. Knowledge of technology grows intrinsically, while humanity grows both intrinsically and extrinsically, depending on knowledge. The hyper-exponential growth model therefore breaks down into two growth models, as described below, one for the Humansphere and one for Knowledge.

## Exponential growth of the *Humansphere*, domesticated land

Drawing inspiration from the "ecological footprint" literature, the stock quantity that includes humanity is modeled as the total human-dominated portion of the global ecosystem, called the *humansphere* [12]. The remaining global biocapacity is the *ecosphere*, representing the portion of the global environment that is not under human domination. *Humansphere* and *ecosphere* are measured in global hectares (gha) and sum to a constant, which is the maximum total biocapacity, $E_0$. Humans are assumed to be a K-selected species as opposed to r-selected [8]. The population of a K-selected species is determined by the carrying capacity, therefore multiplying *humansphere* by the carrying capacity per gha gives the population.

Growth of humanity is modeled as a flow of gha from *ecosphere* to *humansphere*, with value *domestication* = $g$ • *humansphere*. This flow goes to zero as the *ecosphere*, the space into which *humansphere* must expand, goes to zero.

$$g = (I_0 + ln(2)/\tau)(1-exp(u\, p_E)) \tag{7}$$

where $g$ is the *domestication* rate constant measured in reciprocal years (y$^{-1}$), $p_E$ = *ecosphere*/$E_0$, $\tau$ is the initial doubling time of the population in years, and $I_0$ is the initial value of *ignorance*. The negative-valued variable $u$ models the aggressiveness of human growth as *ecosphere* approaches zero, discussed below. A large negative $u$ means aggressive domestication ($u$ is negative so that $exp(u\, p_E)$ is bounded between zero and one.). Flow of *humansphere* back to *ecosphere* is called *rewilding* (see Eq 4). Note that *ignorance*, a number that reflects the mortality rate, initially declines exponentially as *knowledge* takes its place.

## Ecosystem model versus birth/death model

Expressing human population using the total ecological footprint of humanity is functionally equivalent to the more traditional birth/death model. *Rewilding* is equivalent to death, since, upon death, a human's resources are returned to the *ecosphere*. Food that is not eaten counts as carbon sequestered, and waste that is not produced counts as waste assimilated. Therefore,

death converts *humansphere* to *ecosphere*. By the same token, *domestication* is functionally equivalent to birth since it converts *ecosphere* to *humansphere* to support an increase in our numbers.

## Exponential growth of *Knowledge* of technology

*Knowledge* (of technology) flows from *ignorance* (of technology), at the intrinsic rate called *learning* = $\kappa * knowledge$. The optimal value $\kappa$ = 9.6e-3 $y^{-1}$ was determined from the data. The amount of *learning* is the degree to which life expectancy has been increased by science and technology in a given year. Theoretically, maximum *knowledge* implies zero *ignorance*, which unrealistically implies zero death. However, simulations never come close to this value. On the other hand, zero *knowledge* implies a mortality rate of equal to the base value, estimated as $I_0$ = 0.11 $y^{-1}$. Unfortunately $I_0$ could not be precisely fit because it affects the simulations only after the present time. This estimate is derived from the estimated current overshoot of the global biocapacity [40].

## Eventual decline of *Knowledge*

*Knowledge* can and does flow back to *ignorance* in the sense that technology becomes lost or obsolescent. Obsolescence happens when new forms of mortality emerge from advancing technology, such as cancer arising from chemical synthesis of pesticides, draught arising from the efficient depletion of aquifers, and disease transmission arising from the increased ease of long-distance travel. In these cases, innovations that initially increased lifespan later uncovered a cryptic ignorance. To model cryptic ignorance, *knowledge* flows back to *ignorance* with value *obsolescence* = $r * knowledge$, where

$$r = exp(\nu\, p_E) \tag{8}$$

where $p_E$ = *ecosphere*/$E_0$. *Obsolescence* is increased by environmental degradation. The negative-valued variable $\nu$ was fit to late 20th century population data. A larger negative value for $\nu$ leads to less *obsolescence* and steeper population growth.

## Carrying capacity formulation

In this model, the human population is calculated as the amount of *Humansphere* times the carrying capacity for humans, which is the situation for all K-selected species. The equations for carrying capacity were worked out such that they reproduced population growth. To achieve the observed rapid population growth in the 20th century, the exponentially-growing quantity *knowledge* was applied to both population growth and the growth of the carrying capacity (*CC*). In preliminary studies, applying knowledge to only the mortality rate (*rewilding*) or to only the *CC* did not reproduce the observed steepness of 20th century population growth. *CC* is defined as the number of people that can be supported on one global hectare (gha) of *humansphere*. The global biocapacity is treated as an unknown constant value $E_0$, an amount of the world's biocapacity which is split between *ecosphere* and *humansphere* [13]. Both *ecosphere* and *humansphere* contribute resources needed for human life, therefore $CC$ = $cc_E + cc_H$, but only the carrying capacity contributed by the *humansphere* ($cc_H$) is augmented by *knowledge*, which makes logical sense.

$$cc_H = c(1 - exp(d \cdot knowledge))cc_E \tag{9}$$

In this view, the carrying capacity contributed by the *humansphere* is wholly dependent on the *ecosphere* carrying capacity ($cc_E$, Eq 10 below), because food production approaches zero as

ecosystem services approach zero. The loss of ecosystem services that maintain climate stability would leave us with unpredictable temperature, precipitation, and storms. Season-to-season instability and unpredictability limit the efficiency of agriculture. Thus environmental degradation (loss of *ecosphere*) leads to a decreased carrying capacity and therefore a loss of human population. Consider for example that the *ecosphere* includes fresh water aquifers and fossil fuels, factors in food production which are not likely to be replaced by any amount of human invention. *Ecosphere* also encompasses the wild Earth systems and bodies (oceans, atmosphere) that provide ecosystem services and thus define the climate. Loss of *ecosphere* is interpretted to mean climate change, consistent with the role of the ecosystem in maintaining climate. Human food production is also limited by the total area of agricultural lands (*humansphere*) times the maximum carrying capacity of those lands under intense cultivation (***c***) [40]. The term 1-exp(***d*** • *knowledge*) expresses the rise in food production efficiency per gha as *knowledge* increases [6]. Following the "law of diminishing returns" [41], food production rises more slowly with each additional unit of knowledge. The optimal value for the degree of diminishing returns was ***d*** = *-110*.

The optimized model (Table 1) was found to closely reproduce a 65% increase in the total caloric output of agriculture observed during the Green Revolution from 1960–2010 [42, 43]. In the model, the carrying capacity $cc_H$ goes from 0.80 gha per capita in 1960, to 1.33 gha per capita in 2010, a 66% increase. The model also roughly reproduces the total human ecological footprint increase of 225% from 0.75 Earths in 1960 to 1.7 Earths in 2010 [44]. In the World4 simulations, *humansphere* grows 187% over the same period.

## Ecosystem fragility, the parameter *a*

Although carrying capacity is wholly dependent on ecosystem services, those services are related to the biocapacity of the *ecosphere* in a way that cannot be easily assumed. We may hypothesize that damage to the ecosystem is not felt until a certain threshold in degradation is reached. Thereafter the damage to ecosystem services may be rapid and go asymptotically to zero, as described allegorically in Ehrlich's "The Population Explosion" [45] with the "rivet popper" story. Indeed, in many ways, the global ecosystem, like the airplane wing in the story, can take serious damage before it suddenly fails in flight. On the other hand, the environment may be more robust than we know and the *ecosphere* may pass a percentage of its ecosystem services to the *humansphere*, never really collapsing to zero. An asymmetric sigmoid function (Eq 10) is used to express a continuum of unknown non-linear relationships between *ecosphere* and ecosystem services ($cc_E$). The asymmetric sigmoid curve allows us to optimize a single parameter (***a***) to express the degree of robustness or fragility of the ecosystem. If ***a*** is low, the environment responds robustly to depletion of *ecosphere* by retaining ecosystem services within the *humansphere*, whereas if ***a*** is high then the ecosystem is fragile and critical services are lost suddenly, as described in Ehrlich's story. This function allows to explore ecosystem fragility by fitting proposed non-linear ecosystem behavior to population.

$$cc_E = \boldsymbol{b}\, p_E^{(0.5/(1+p_E-2\boldsymbol{a}))} \tag{10}$$

where $p_E$ = *ecosphere*/***E_0***. Fig 5 shows the shape of the $cc_E$ function and how it changes with ***a***.

Variable ***a*** was initially set to a value that places the global ecosystem today at about one-half *humansphere* and one-half *ecosphere* in 2005, the year of "peak oil" [46], albeit that peak date should perhaps be pushed forward by new discoveries of natural gas and new mining technologies. Fossil fuel is a dominant resource in raising the carrying capacity in the 20th century and its numbers are well studied and readily available, however it should be recognized that other limiting natural resources and ecosystem services may "run out" before fossil fuel

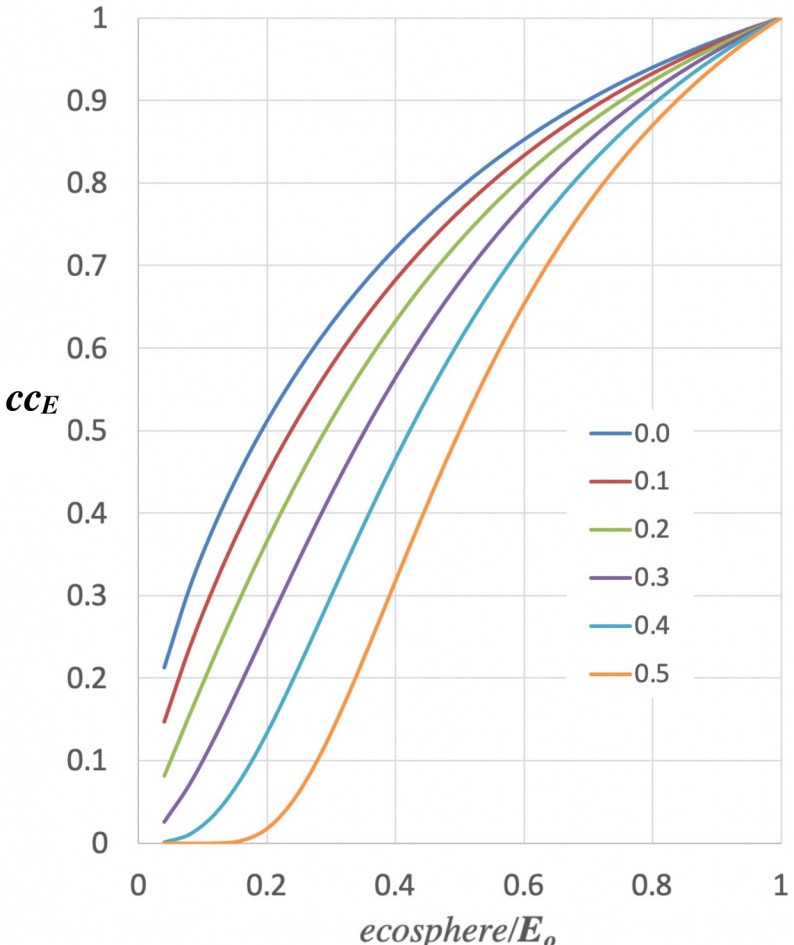

**Fig 5. Plot of Eq 10, the *ecosphere* component of the carrying capacity for humans.** Increasing values of *a* move the curve to the right, meaning ecosystem services are more fragile with respect to the *ecosphere*.

does. Therefore, the variable *a* was set, not by *a priori*, but by fitting to late 20th century population data as described below. This variable, along with $E_0$, is one of the most critical to the accurate fitting of late 20th century, post-hyperexponential growth. The optimal value of *a* was found to be in a range from 0.27 to 0.48 depending other variables. Fig 3A–3C shows the shape of the range of optimality in dimensions *a*, $E_0$, and *v*.

## Model fitting to population data

Variable setting were determined by least-squares fitting using the program *Hyperfit*. All variables in **bold italics** in Table 1 were fit to data. The most accurate and reliable data relevant to human population is the population itself. Global population numbers have been estimated for as far back as 10,000 years ago, and recent numbers are likely to be correct within 3% [28]. Exploratory curve fitting was used to test functional forms using Microsoft Excel, using the Solver plug-in [47]. Other global numbers, such as the ecological footprint, world forest cover, atmospheric carbon, global economic output, etc. were used in supporting roles only.

Variables were solved in three stages. First, intrinsic growth parameters ($H_0$, $K_0$, $\tau$, and $\kappa$) were fit to years 1–1970. Second, the three "footprint" variables that affect the balance between

normal exponential growth and hyper-exponential growth (**b, c** and **d**) were fit to years 1900–2010. Third, the late 20th century discrepancy (1970–2010) was reconciled using the variables (**E₀, a, u, v**) that determine the total global biocapacity, the fragility of ecosystem services, and the strength of feedback from the environment on the carrying capacity for humans. One variable, the baseline mortality rate in Year 0 (**I₀**) does not have an effect on population until after the present date, and therefore could not be fit. **I₀** may be arbitratilly set to 0.11 y$^{-1}$ to reproduce previously estimated overshoot values (1.7 earths [44]). With this setting, the current population is 1.7 times the equilibrium value, which would be around 4 billion in 2100 using **I₀** = 0.11. Also, four variables (**w, p, s_y, p_y**) were created for the hypothetical implementation of environmental conservation policy in the 21st century.

### *Hyperfit*, a multivariable sampling algorithm

Finally, a range of least-squares minima was identified by exhaustive sampling of variables within the ranges defined by the heirachical fit method, using *Hyperfit*. *Hyperfit* is a program in Fortran90 that runs the World4 model for any number of ranges of variables, either randomly or exhaustively. In exhaustive sampling mode, *Hyperfit* reads an input file containing the ranges to be sampled and parameters for the range of years to be fit and the sampling density. The output is a matrix of mean square residuals summed over the range of years sampled. Up to 4 variables can be sampled exhaustively. The program outputs to the plotting program *gnuplot* [50]. In random sampling mode, *Hyperfit* samples all variables within ranges defined in the input file using a flat probability distribution, then runs a simulation and saves the parameters if the residual is below a cutoff (*e.g.* 0.5e8 for the years 1970–2010). The set of best fit parameters were passed back into *Hyperfit* to generate trajectories for plotting, using Microsoft Excel.

## Model availability

The interactive model World4 may be accessed on the InsightMaker web site [48]. By cloning the model, anyone can make changes. Complete parameters and equations for the model are included in the model itself and in Table 1. The program *Hyperfit* is freely available from the author upon request, or from OSF [49]. *Hyperfit* requires a fortran95 compiler and installation of *gnuplot* [50].

## Postscript

During review of this paper, the first results of the global 2020 censuses began to come out. The US population grew by 7.4% from 2010 to 2020, much slower than the 9.7% growth observed between the 2000 and 2010 censuses. This implies, if the US population remains at 4.44% of the world population, that the world population last year was 7.45 billion, which falls within the range predicted by World4, 7.2 to 7.6 billion, but is well below the projections from the United Nations model, whose site reports a population of 7.79 billion for 2020. Global population numbers have still not been released as of this date.

## Acknowledgments

Several people have provided feedback on this manuscript in its current form or in one of its earlier forms, including Robert Wyman, George Richardson, and Lee Badger. Special thanks to the students who took the RPI computational biology course Human Population and tested several alternative models as part of their homework assignments.

## Author Contributions

**Conceptualization:** Christopher Bystroff.

**Data curation:** Christopher Bystroff.

**Formal analysis:** Christopher Bystroff.

**Funding acquisition:** Christopher Bystroff.

**Investigation:** Christopher Bystroff.

**Methodology:** Christopher Bystroff.

**Software:** Christopher Bystroff.

**Validation:** Christopher Bystroff.

**Visualization:** Christopher Bystroff.

**Writing – original draft:** Christopher Bystroff.

**Writing – review & editing:** Christopher Bystroff.

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
