## [Decision Letter · Decision Letter 0]

10 Mar 2021

PONE-D-21-03679

Footprints to Singularity: A global population model explains late 20th century slow-down and predicts peak within ten years.

PLOS ONE

Dear Dr. Bystroff,

Thank you for submitting your manuscript to PLOS ONE. After careful consideration, we feel that it has merit but does not fully meet PLOS ONE’s publication criteria as it currently stands. Therefore, we invite you to submit a revised version of the manuscript that addresses the points raised during the review process.

The reviewers agreed on major revision; and the points raised in the reviewers' reports are centred on two major issues: Poor presentation of the data, model and analysis as stated by Reviewer 1; and that the literature is underrepresented and the contributions to the literature are not clearly explained as argued by Reviewer 2. We expect that you can address these issues in the revisions and your revised manuscript will be reconsidered accordingly.

We look forward to receiving your revised manuscript.

Kind regards,

Eda Ustaoglu, PhD

Academic Editor

PLOS ONE

Journal Requirements:

3. Thank you for stating in your financial disclosure: 

'C.B. was sopprted by an award from a private foundation that wishes to remain anonymous. The funders had no role in study design, data collection and analysis, decision to publish, or preparation of the manuscript.i

PLOS ONE requires you to include in your manuscript further information about the funder so that any relevant competing interests can be assessed. Please respond to the following questions:

Please state whether any of the research costs or authors' salaries were funded, in whole or in part, by a tobacco company (our policy on tobacco funding is at http://journals.plos.org/plosone/s/disclosure-of-funding-sources) Please state whether the donor has any competing interests in relation to this work (see http://journals.plos.org/plosone/s/competing-interests) .Please state whether the identity of the donor might be considered relevant to editors or reviewers’ assessment of the validity of the work.If the donors have no perceived or actual competing interests, please state: “The authors are not aware of any competing interests”.

This information should be included in your cover letter. We will amend your financial disclosure and competing interests on your behalf.

Reviewers' comments:

Reviewer's Responses to Questions

**Comments to the Author**

1. Is the manuscript technically sound, and do the data support the conclusions?

Reviewer #1: Partly

Reviewer #2: Partly

2. Has the statistical analysis been performed appropriately and rigorously? 

Reviewer #1: N/A

Reviewer #2: N/A

3. Have the authors made all data underlying the findings in their manuscript fully available?

Reviewer #1: No

Reviewer #2: Yes

4. Is the manuscript presented in an intelligible fashion and written in standard English?

Reviewer #1: No

Reviewer #2: Yes

5. Review Comments to the Author

Reviewer #1: I have given detailed comments on the paper in the attached pdf. But my overall comments are as follows.

The topic of the paper is of considerable interest.

But it is very poorly presented with material all over the place.

The mathematical model is poorly presented

The analysis of the parameter fitting is not at all clear.

The assumptions of the work are not clearly articulated.

The data that the model is based on is not clear.

I did not enjoy reading this paper as a mathematical modeller as the presentation was so disjointed.

The work needs a complete overhaul. If that is done successfully then it could well be suitable for publication. See my comments in the attached pdf.

The paper is very close to being rejected outright but the topic is a timely one. However, I do not want to see it again. I was very disappointed with the way the modelling was presented and the disjointed nature of the work..

Reviewer #2: This is a fairly standard variation of the 'world model' of the Club of Rome flavor and its may variants over the years.

The notion of 'knowledge and obsolescence' may be slightly novel, but if they are modeled as 'stocks' as they are here, this is no different than a standard economic growth model with investment and depreciation. The mathematical symbols have just been given different names.

The manuscript misses a very large literature on economic growth and the environment that uses coupled differential equation models (i.e. system dynamics) developed in mathematical bioeconomics. This is not surprising given the 'engineering-system dynamics' tradition from whence this model comes. These two literatures have often missed one another, creating useless discussions (i.e. the conflict/arguments between the Club of Rome lead by D. H. and D.L. Meadows and multiple economists who focus on economic growth such as Robert Solow, Geoffrey Heal, and Joseph Stiglitz in the 1970's and 80's) and multiple re-inventions of the wheel.

To be publishable, this manuscript must demonstrate what it adds to this literature. The trajectories shown in Figures 1-4 are very generic - you can find countless versions of these pictures in the literature. Fitting them to historical data is not difficult or particularly interesting. Thus, I suggest the authors carefully articulate what their model adds, taking into account the literature they missed, including

1) Brander, James A., and M. Scott Taylor. "The simple economics of Easter Island: A Ricardo-Malthus model of renewable resource use." American economic review (1998): 119-138.

There are many variations on this. It is essentially a 'world model'.

2) Anderies, John M. "Economic development, demographics, and renewable resources: a dynamical systems approach." Environment and Development Economics (2003): 219-246.

Again a world model. This one rigorously analyses the 'generic' behavior of all such models. You will see all the trajectories there that appear in this manuscript.

3) Beltratti, Andrea. "Growth with natural and environmental resources." New directions in the economic theory of the environment 25 (1997): 7.

4) Beltratti, Andrea. Models of economic growth with environmental assets. Vol. 8. Springer Science & Business Media, 1996.

5) Beltratti, Andrea, Geoffrey M. Heal, and Graciela Chichilnisky. "Sustainable growth and the green golden rule." Available at SSRN 1374662 (1995).

6) Chichilnisky, Graciela, Geoffrey Heal, and Andrea Beltratti. "The green golden rule." Economics Letters 49.2 (1995): 175-179.

7) Xepapadeas, Anastasios. "Economic growth and the environment." Handbook of environmental economics 3 (2005): 1219-1271.

and follow up on the many other models cited in those papers.

Until the model is correctly placed in the existing literature and its novel contribution clarified, it is just another variation of hundreds of such models.

6. PLOS authors have the option to publish the peer review history of their article (what does this mean?). If published, this will include your full peer review and any attached files.

Reviewer #1: No

Reviewer #2: No

---

## [Decision Letter · Decision Letter 1]

28 Apr 2021

PONE-D-21-03679R1

Footprints to Singularity: A global population model explains late 20th century slow-down and predicts peak within ten years.

PLOS ONE

Dear Dr. Bystroff,

Thank you for submitting your manuscript to PLOS ONE. After careful consideration, we feel that it has merit but does not fully meet PLOS ONE’s publication criteria as it currently stands. Therefore, we invite you to submit a revised version of the manuscript that addresses the points raised during the review process.

Please add the postscript to the paper.

We look forward to receiving your revised manuscript.

Kind regards,

Eda Ustaoglu, PhD

Academic Editor

PLOS ONE

Journal Requirements:

Additional Editor Comments (if provided):

Dear Dr Bystroff,

regarding your request to add a postscript in the paper, you can do this now.

I submit 'minor revision decision' to allow you adding the postscript.

Reviewers' comments:

Reviewer's Responses to Questions

**Comments to the Author**

1. If the authors have adequately addressed your comments raised in a previous round of review and you feel that this manuscript is now acceptable for publication, you may indicate that here to bypass the “Comments to the Author” section, enter your conflict of interest statement in the “Confidential to Editor” section, and submit your "Accept" recommendation.

Reviewer #2: All comments have been addressed

2. Is the manuscript technically sound, and do the data support the conclusions?

Reviewer #2: (No Response)

3. Has the statistical analysis been performed appropriately and rigorously? 

Reviewer #2: (No Response)

4. Have the authors made all data underlying the findings in their manuscript fully available?

Reviewer #2: (No Response)

5. Is the manuscript presented in an intelligible fashion and written in standard English?

Reviewer #2: (No Response)

6. Review Comments to the Author

Reviewer #2: (No Response)

7. PLOS authors have the option to publish the peer review history of their article (what does this mean?). If published, this will include your full peer review and any attached files.

Reviewer #2: No

---

## [Author Response · Author response to Decision Letter 1]

3 May 2021

Reviewer #2: All comments have been addressed

Author response:

Thank you!

The revised manuscript contains a postscript, located at the end before References. In the postscript I acknowledge the release of limited 2020 census data and not that it agrees with my predictions.

---

## [Editor Report · Decision Letter 2]

6 May 2021

Footprints to Singularity: A global population model explains late 20th century slow-down and predicts peak within ten years.

PONE-D-21-03679R2

Dear Dr. Bystroff,

We’re pleased to inform you that your manuscript has been judged scientifically suitable for publication and will be formally accepted for publication once it meets all outstanding technical requirements.

Kind regards,

Eda Ustaoglu, PhD

Academic Editor

PLOS ONE
---

## [Editor Report · Acceptance letter]

10 May 2021

PONE-D-21-03679R2 

Footprints to Singularity: A global population model explains late 20th century slow-down and predicts peak within ten years. 

Dear Dr. Bystroff:

I'm pleased to inform you that your manuscript has been deemed suitable for publication in PLOS ONE. Congratulations! Your manuscript is now with our production department. 

Kind regards, 

on behalf of

Dr. Eda Ustaoglu 

Academic Editor

PLOS ONE